# Combined Effects of Metabolic Abnormalities and Obesity on Cardiovascular Diseases among Korean Postmenopausal Women

**DOI:** 10.3390/healthcare9081064

**Published:** 2021-08-19

**Authors:** Jin Suk Ra

**Affiliations:** College of Nursing, Chungnam National University, Daejeon 35015, Korea; jinsukra@cnu.ac.kr; Tel.: +82-42-538-8333

**Keywords:** cardiovascular diseases, metabolic syndrome, obesity, postmenopausal

## Abstract

Combined effects of metabolic abnormalities, including metabolic syndrome and obesity, should be identified to screen postmenopausal women at risk of developing cardiovascular diseases. The purpose of this study was to identify the combined effects of metabolic abnormalities and obesity on cardiovascular diseases among postmenopausal Korean women (aged 40–83 years). Data of 5959 postmenopausal women from the Korean National Health and Nutrition Examination Survey (2015–2018) were secondarily analyzed. Using complex simple analysis procedures, logistic regression analysis was performed to identify the combined effect of metabolic abnormalities and obesity on cardiovascular diseases among postmenopausal Korean women. In combination, metabolic syndrome (more than three metabolic abnormalities) and obesity (overweight [≥23 kg/m^2^ and <25 kg/m^2^ in body mass index] and obese [>25 kg/m^2^ in body mass index]) increased the likelihood of developing cardiovascular diseases but combining more than one metabolic abnormality and obesity did not. Combining metabolic syndrome and non-obesity (underweight and normal weight) increased the likelihood of the prevalence of cardiovascular diseases but combining more than one metabolic abnormality and non-obesity did not. Increased cardiovascular diseases in postmenopausal women may be more commonly associated with metabolic syndrome having multiple metabolic abnormalities, but not obesity. Thus, instead of simple weight control, early management of metabolic syndrome is recommended to prevent cardiovascular disease among postmenopausal Korean women.

## 1. Introduction

Postmenopausal women tend to experience increases in body weight and abdominal obesity, which result from estrogen depletion, decreased physical activity, and increased sedentary behaviors [1,2]. Obesity and abdominal visceral fat accumulation in the postmenopausal period lead to an increased risk of metabolic abnormalities including insulin resistance, hypertension, low high-density lipoprotein (HDL) cholesterol levels, and high levels of low-density lipoprotein (LDL) cholesterol and triglycerides (TG) [3,4]. According to previous studies, obesity and metabolic syndrome with the clustering of three or more metabolic abnormalities are an independent risk factor for the development of cardiovascular diseases (CVDs) and consequent deaths [5]. Thus, early management of obesity and metabolic abnormalities may be important for the prevention of CVDs in postmenopausal women. In view of the interactive effects of combined patterns of metabolic abnormalities and obesity, the combined effects of metabolic abnormalities including metabolic syndrome and obesity should be studied; this will facilitate screening of postmenopausal women at risk, who primarily need management for the prevention of CVDs [5,6,7].

In this context, previous studies attempted to identify the risk of CVDs according to the combined patterns of metabolic abnormalities and obesity (overweight and obesity) (e.g., with/without metabolic syndrome and obesity, and with/without metabolic syndrome and non-obesity) [7]. However, the significance of the combined effects of metabolic abnormalities and obesity on CVDs and the associated complications (e.g., mortality) were unclear. In a study on women aged 20 to 80 years, normal weight women with metabolic syndrome (more than three metabolic abnormalities) showed an increased risk of CVDs, while obese women without metabolic syndrome (less than two metabolic abnormalities) did not [8]. In a study on women with a mean age of 45.7 years, obese women without any metabolic abnormalities did not show a significantly increased risk of ischemic heart disease [9]. In addition, according to a meta-analysis [7], obesity without metabolic syndrome was also associated with a significantly increased risk of CVDs. However, a study showed that obesity with at least one metabolic abnormality and normal weight with more than one metabolic abnormality conferred an increased risk of CVDs during a 5-year follow up [9]. The unclear significance of the combined effects of metabolic abnormalities and obesity on CVDs were attributed to differences in the sex and ethnicity of samples, and the application of different criteria for metabolic disease [9,10,11,12]. Thus, it is necessary to identify the combined effects of metabolic abnormalities and obesity, considering the criteria for metabolic disease, for the prevention of CVDs among postmenopausal Korean women. However, few studies have considered the sex and criteria of metabolic disease in the samples. In this context, the purpose of this study was to identify the combined effect of metabolic abnormalities and obesity on CVDs in postmenopausal Korean women according to two criteria for metabolic disease (more than one metabolic abnormality and more than three metabolic abnormalities, namely, metabolic syndrome).

CVDs may develop as a result of interaction between various biological, psychological, and social determinants [13]. According to the biopsychosocial holistic model of cardiovascular health, interactive factors such as biological factors (e.g., age, sex, and genomics), social factors (e.g., socioeconomic status), and psychological factors (e.g., stress, depression, and health behaviors) influence cardiovascular health [14]. Thus, based on a literature review, biological factors including age, family history of hypertension, dyslipidemia, and type 2 diabetes mellitus (T2DM) would have an influence as covariates [13,15]. Among social factors, family socioeconomic status and educational level have been identified [13,15]. In addition, stress, depression, current or past smoking experiences, current binge alcohol consumption, physical activity, and sedentary behaviors are notable psychological factors that qualify as covariates [13,15]. Thus, this study was performed to identify the combined effect of metabolic abnormalities and obesity on CVDs among postmenopausal Korean women after controlling for the known covariates.

## 2. Materials and Methods

### 2.1. Study Design and Samples

This cross-sectional study analyzed secondary data from the Korean National Health and Nutrition Examination Survey between 2015–2018. Among the 21,578 women who participated in the survey, 13,049 women aged over 40 years were selected. Among the selected women, 7092 women who were either premenopausal or did not complete the questionnaire that assessed health and nutritional status were excluded. Finally, 5957 women (aged 40–83 years) were included in this study for data analysis. 

### 2.2. Measurements

#### 2.2.1. Outcome Variable

##### Cardiovascular Diseases

CVDs were evaluated using a single question to identify a diagnosis of coronary heart disease (angina pectoris and myocardial infarction) and cerebrovascular disease (stroke) by a physician specializing in CVDs.

#### 2.2.2. Independent Variable

##### Metabolic Abnormalities

The considered metabolic abnormalities included abdominal obesity, high blood pressure, low HDL cholesterol, high triglycerides, and high fasting glucose. In addition, metabolic syndrome was defined by the presence of three or more metabolic abnormalities based on the criteria proposed by the American Heart Association and the National Heart, Lung, and Blood Institute [16]. The exception was abdominal obesity, for which the criteria proposed by the Korean Society for the Study of Obesity were applied [17].

Specific criteria for the five metabolic abnormalities were as follows: (i) abdominal obesity: ≥85 cm waist circumference; (ii) high blood pressure: ≥130/85 mmHg or intake of medication for the management of hypertension; (iii) low HDL cholesterol: <50 mg/dL or intake of medication for the management of low HDL; (iv) high triglycerides: ≥150 mg/dL or intake of medication for the management of high triglycerides, and (v) high fasting glucose: ≥100 mg/dL or intake of medication for the management of hyperglycemia.

##### Adiposity

Adiposity was evaluated using body mass index (BMI), calculated with objectively measured height and weight. The BMI was then categorized into underweight (<18.5 kg/m^2^); normal weight (≥18.5 kg/m^2^ and <23 kg/m^2^); overweight (≥23 kg/m^2^ and <25 kg/m^2^), and obese (>25 kg/m^2^) categories [18]. Underweight and normal weight categories were classified as non-obesity, and overweight and obesity were classified as obesity. 

#### 2.2.3. Covariates

##### Biological Factors

The categories for age were 40–64 years and ≥65 years. A family history of hypertension, dyslipidemia, T2DM, or CVDs was assessed using a single question on the history of hypertension, dyslipidemia, T2DM, or CVDs in direct family members. Responses included either a “yes” or “no.” 

##### Social Factors

The socioeconomic status of the family was assessed based on their perceived socioeconomic status. Responses were categorized as either high, middle, or low. Educational level was assessed based on their highest level of educational attainment. Responses were categorized as follows: less than graduation from elementary school, graduation from middle school, graduation from high school, and more than graduation from college.

##### Psychological Factors

Stress was assessed using a single question on the level of daily stress. The responses were categorized as either severe, little, or not at all. Depression was assessed based on whether they were diagnosed by a psychiatrist; responses were recorded as either “yes” or “no.” Current or past smoking history was assessed using a single question regarding current or past smoking experiences; responses were recorded as either “yes” or “no.” Current binge alcohol consumption was assessed based on the following: amount and frequency of alcohol use and frequency of heavy drinking using alcohol consumption questions (AUDIT-C). The total scores ranged from 0 to 12 points, with 0 to 4 points for the response rate of each item. More than six points were considered as binge alcohol consumption, based on the criteria proposed by Woo et al. [19]. Physical activity was assessed using the Metabolic Equivalent Task (MET)-minutes of Global Physical Activity Questionnaire analysis guide (version 2.0) [20]. Physical activity included activities during work, travel to and from places, and leisure. This was calculated as the sum of walking or cycling, and moderate and vigorous MET minutes in a week. The level of physical activity was classified into moderate to vigorous intensity (≥600 MET-min per week) and low intensity (<600 MET-min per week) categories. Sedentary behavior was assessed using the question: “How many hours do you usually sit or lie down a day?” Responses were categorized into <8 h a day and ≥8 h a day, and sedentary time with more than 8 h a day was considered as prolonged sedentary behavior according to the guidelines of the Department of Health & Social Care [21]. 

### 2.3. Ethical Consideration

This study was exempted from institutional review board (IRB) review, as it was a secondary analysis (approval no. 202105-SB-063-01).

### 2.4. Statistical Analysis

Following guidelines proposed by the Korean National Health and Nutrition Examination Survey, data analyses were performed by complex simple analysis procedures using SPSS version 26.0 (IBM, Armonk, NY, USA). The prevalence of CVDs, metabolic abnormalities, adiposity, and covariates (biological, social, and psychological factors) were analyzed as frequencies and percentages. Logistic regression analysis was performed with or without controlling the covariates, to identify the combined effect of metabolic abnormalities and obesity on CVDs among Korean postmenopausal women.

## 3. Results

### 3.1. Characteristics of Cardiovascular Diseases, Metabolic Abnormalities, Adiposity, and Covariates among Participants

Overall, 1.6% of the postmenopausal women had CVDs; 50.6% of the participants had at least 1 metabolic abnormality, and 14.5% had metabolic syndrome with at least three metabolic abnormalities. In addition, 41% of the participants were overweight and obese. Among the participants who were underweight and normal weight, 32.2% of participants had at least one metabolic abnormality, and 3.5% had metabolic syndrome with at least three metabolic abnormalities. Among the participants who were overweight and obese, 77.1% of participants had at least one metabolic abnormality, and 30.3% had metabolic syndrome with at least three metabolic abnormalities. The prevalence of the covariates (biological, social, and psychological factors) has been presented in Table 1.

### 3.2. Combined Effects of Metabolic Abnormalities and Obesity among Korean Postmenopausal Women

Without controlling for covariates, participants with more than one metabolic abnormality and obesity combined had an increased likelihood of developing CVDs (crude odds ratio [OR]: 4.43, 95% confidence interval [CI]: 1.63–12.09). Combined metabolic syndrome and obesity increased the likelihood of developing CVDs (crude OR: 7.00, 95% CI: 3.28–14.94); combined metabolic syndrome and non-obesity (crude OR: 10.53, 95% CI: 3.65–30.33) was associated with an increased likelihood of developing CVDs, but a combination of more than one metabolic abnormality and non-obesity (crude OR: 3.05, 95% CI: 0.98–9.45) was not. Without any metabolic abnormalities or metabolic syndrome (less than two metabolic abnormalities), obese participants did not show an increased likelihood of developing CVDs (Table 2). 

On controlling for covariates, combined metabolic syndrome and obesity was associated with an increased likelihood of developing CVDs (adjusted OR: 2.67, 95% CI: 1.27–5.62), but combining more than one metabolic abnormality and obesity (adjusted OR: 1.29, 95% CI: 0.47–3.53) was not. In addition, combining metabolic syndrome and non-obesity was associated with an increased likelihood of developing CVDs (adjusted OR: 3.09, 95% CI: 1.09–8.78), but more than one metabolic abnormality and non-obesity (adjusted OR: 0.95, 95% CI: 0.31–2.95) was not. In addition, obese participants without metabolic abnormalities did not show an increased likelihood of developing CVDs (Table 2).

## 4. Discussion

This study identified the combined effects of metabolic abnormalities and obesity on CVDs among postmenopausal Korean women. The results of this study suggest that the association between CVDs among Korean postmenopausal women and metabolic syndrome with more than three metabolic abnormalities may be stronger than that with more than one metabolic abnormality and obesity. 

In previous experimental and observational studies, estrogen demonstrated a protective effect on CVDs in women [22]. Thus, postmenopausal women may have an increased risk of CVDs than men of the same age owing to estrogen deficiency associated with the loss of ovarian function; however, premenopausal women are considered to be at less risk of CVDs than men of the same age [23]. In this context, deficiency of estrogens may result in increased abdominal obesity or accumulation of abdominal visceral fat, which is linked to Type 2 DM with increased insulin resistance, dyslipidemia including reduced HDL cholesterol levels and hypertriglyceridemia, and hypertension in postmenopausal women, including postmenopausal Korean women [24,25,26,27]. In view of the strong association between metabolic abnormalities (e.g., metabolic syndrome) and CVDs during menopausal transition periods [3], increased abdominal obesity in the postmenopausal period may confer a greater risk of developing. In this context, abdominal obesity rather than BMI was a more consistent and stronger predictor of coronary heart disease in a previous study including 24,508 individuals aged 45–79 years [8,28]. Cho et al. [29] also reported that abdominal obesity, not overall obesity, was significantly associated with obstructive coronary artery diseases in postmenopausal Korean women. Obese women without metabolic abnormalities may have a relatively low risk of abdominal obesity; conversely, metabolic syndrome with obesity may confer a higher risk of abdominal obesity in women. In this context, studies have shown that metabolic abnormities including metabolic syndrome are stronger predictors of CVDs than obesity [8,10]. According to a meta-analysis of prospective cohort studies and a community-based study including 6215 individuals with metabolic syndrome, individuals with normal weight were also at increased risk of developing CVDs (coronary artery diseases and stroke) [7,30]. However, a 10-year follow-up study including 3189 obese individuals without metabolic abnormalities did not report an increased risk of CVDs [9]. Thus, for the prevention of CVDs among postmenopausal Korean women, prevention and management of metabolic abnormalities may be more relevant than simply maintaining a normal weight. 

In a study including White, Black, and Hispanic postmenopausal overweight and obese women, less than one metabolic abnormality was not found to be associated with an increased risk of CVDs, regardless of race and ethnicity [10]. In addition, a previous study on women using Korean national data showed that more than one metabolic abnormality was not associated with cardiovascular mortality, regardless of adiposity [5]. Thus, postmenopausal Korean women with a greater number of metabolic abnormalities, such as metabolic syndrome, may have a higher risk of CVDs. Thus, early management of metabolic abnormalities, prior to the development of metabolic syndrome, is recommended to prevent CVD among postmenopausal Korean women. 

Based on big national data, this study identified that metabolic syndrome rather than obesity should be more urgently managed for the prevention of CVDs among postmenopausal Korean women. However, there were limitations to this study. First, it employed a cross-sectional study design, and verification of causal relationships between independent and outcome variables was limited; longitudinal cohort studies are therefore necessary to address this issue. Second, limited covariates were included for secondary data analysis. Third, this study included only Korean women; however, ethnic differences are known to influence the impact of metabolic abnormalities on CVDs, and Korea has an increasingly multiethnic society. Thus, further studies are needed to identify ethnic differences in the influence of metabolic abnormalities on CVDs among multiethnic cohorts. Fourth, although biological, social, and psychological factors were controlled as covariates, studies comparing the effect of metabolic abnormalities and obesity on CVDs according to the level of biological, social, and psychological factors (e.g., age) are required. Fifth, this study focused on the combined effects of metabolic abnormalities and obesity on CVDs. However, with the significant association, metabolic abnormalities and obesity might have an interactive effect on CVDs. Thus, further studies are needed to test the interactive effect between metabolic abnormalities and obesity on CVDs. Sixth, this study did not identify key metabolic abnormalities associated with CVDs, but only confirmed the significant effect of metabolic syndrome on developing CVDs in postmenopausal Korean women. Thus, further studies focusing on the verification of key metabolic abnormalities associated with developing CVDs among postmenopausal Korean women are required. 

## 5. Conclusions

The results of this study suggest that obesity may not be a significant factor associated with CVDs among postmenopausal Korean women. The increased risk of CVDs in this population may be associated more with metabolic syndrome consequent to multiple cumulative metabolic abnormalities. Thus, early management is primarily recommended in postmenopausal Korean women developing metabolic syndrome, to prevent cardiovascular disease. Healthcare providers should focus on prevention strategies for metabolic abnormalities, including the management of abdominal obesity, and not merely on weight loss. 

## Figures and Tables

**Table 1 healthcare-09-01064-t001:** Characteristics of cardiovascular diseases, metabolic abnormalities, adiposity, and covariates among the participants.

Variables	Categories	*n* (%)
Cardiovascular Diseases	Yes	106 (1.6)
	No	5853 (98.4)
Metabolic abnormalities		2753 (49.4)
More than one metabolic abnormality	Yes	3206 (50.6)
	No	
Metabolic syndrome (More than three metabolic abnormalities)	Yes	942 (14.5)
	No	5017 (85.5)
Components of metabolic abnormalities		
Abdominal obesity	Yes	1148 (17.9)
	No	4811 (82.1)
High blood pressure	Yes	828 (12.3)
	No	5131 (87.7)
Low high density lipoprotein cholesterol	Yes	1913 (30.5)
	No	4046 (69.5)
High triglycerides	Yes	1259 (19.5)
	No	4700 (80.5)
High fasting glucose	Yes	1295 (19.9)
	No	4664 (80.1)
Adiposity		
Underweight		362 (6.7)
Normal weight		3021 (52.3)
Overweight		1110 (17.7)
Obesity		1466 (23.3)
Metabolic abnormalities according to adiposity		
Underweight and normal weight (*n* = 3383)	More than one metabolic abnormality	1185(32.2)
	Abdominal obesity	23 (0.6)
	High blood pressure	254 (6.4)
	Low high density lipoprotein cholesterol	724 (19.9)
	High triglycerides	365 (10.0)
	High fasting glucose	411 (10.6)
	Metabolic syndrome	136 (3.5)
Overweight and obesity (*n* = 2576)	More than one metabolic abnormality	2021 (77.1)
	Abdominal obesity	1125 (42.9)
	High blood pressure	574 (20.6)
	Low high density lipoprotein cholesterol	1189 (45.7)
	High triglycerides	894 (33.2)
	High fasting glucose	884 (33.5)
	Metabolic syndrome	806 (30.3)
Biological factors		
Age (years)	40–64	3460 (52.1)
	≥65	2499 (47.9)
Family history of hypertension, dyslipidemia, type 2 diabetes mellitus, cerebrocardiovascular disease	Yes	3835 (62.7)
	No	2124 (37.3)
Social factors		
Socioeconomic status of family	High	2540 (42.3)
	Middle	1212 (20.0)
	Low	2207 (37.7)
Education level	Less than graduation of middle school	844 (12.3)
	Graduation of high school	2288 (39.5)
	More than graduation of college	2827 (48.2)
Psychological factors		
Stress	Severe	1809 (31.7)
	A little	3515 (58.2)
	Almost none	635 (10.1)
Depression	Yes	269 (4.6)
	No	5690 (95.4)
Current or past smoking history	Current and past smokers	844 (14.9)
	Non-smokers	5115 (85.1)
Current binge alcohol consumption	Almost every day	114 (1.9)
	Once a week	536 (9.3)
	Less than once a month	2194 (39.1)
	Never	3115 (49.7)
Physical activity	Moderate to vigorous intensity	3918 (66.7)
	Low intensity	2041(33.3)
Sedentary behaviors (hours a day)	≥8	3185 (55.0)
	<8	3184 (48.1)

*Note 1.* Number was underweighted and percentage was weighted.

**Table 2 healthcare-09-01064-t002:** Combined effects of metabolic abnormalities and obesity among Korean postmenopausal women.

Variables	Cardiovascular Diseases	
	Crude OR (95% CI)	Adjusted OR (95% CI)
***Without metabolic abnormalities and non-obesity*** **(Reference)**	1	1
With metabolic abnormalities and obesity		
More than one metabolic abnormality	4.43 (1.63–12.09)	1.29 (0.47–3.53)
Metabolic syndrome	7.00 (3.28–14.94)	2.67 (1.27–5.62)
With metabolic abnormalities and non-obesity		
More than one metabolic abnormality	3.05 (0.98–9.45)	0.95 (0.31–2.95)
Metabolic syndrome	10.53 (3.65–30.33)	3.09 (1.09–8.78)
Without metabolic abnormalities and overweight/obesity		
Without metabolic abnormalities	1.87 (0.36–9.69)	1.00 (0.26–3.80)
Without metabolic syndrome (Less than two metabolic abnormalities)	1.41 (0.59–3.36)	0.92 (0.41–2.06)

*Note 1:* OR = odds ratio; CI = confidence interval. *Note 2:* adjusted OR = adjusted for covariates (biological, social, and psychological factors).

## Data Availability

Data were acquired from KDCA and are available from https://knhanes.kdca.go.kr/knhanes/main.do (accessed on 6 May 2021).

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
