# Peer review of "Combined Effects of Metabolic Abnormalities and Obesity on Cardiovascular Diseases among Korean Postmenopausal Women"

_healthcare, 2021, doi:10.3390/healthcare9081064_

Round 1

Reviewer 1 Report

The paper is a classical cross-sectional study on the association between metablic abnormalities and occurence of cardiovascular diseases. The results seem to be reseasonable. However, I do have some suggestions to offer.

First, when considering the effect of metabolic abnormalities and obesity on cardiovascular diseases (CVD), I suggest further analyses of interaction between metabolic abnormalities and obestiy on CVD.

Second, did the authors ask the subjects about usage of medication against metablic syndromes such as antihypertensives, oral diabetes drugs or insulins? These need to be included as covariates in analyses, if not described as limitations.

Author Response

Title: Combined effects of metabolic abnormalities and obesity on cardiovascular diseases among Korean postmenopausal women

We appreciate the reviewers for your valuable comments. The following are our responses to each comment. To make it easier for you to follow how changes were made, our changes appear in red ink in the revised text.

Reviewer 1.

The paper is a classical cross-sectional study on the association between metablic abnormalities and occurrence of cardiovascular diseases. The results seem to be reasonable. However, I do have some suggestions to offer.

  1. First, when considering the effect of metabolic abnormalities and obesity on cardiovascular diseases (CVD), I suggest further analyses of interaction between metabolic abnormalities and obesity on CVD.

Response: I agree with the need for further analysis. However, purpose of this study was to identify combined effect of metabolic abnormalities and obesity on CVDs. If I want test interaction between metabolic abnormalities and obesity, further study should be performed with another purpose and hypothesis. Thus, in limitation section, the need of further studies regarding interaction between  metabolic abnormalities and obesity was described (pages 8, line 251~253).

  1. Second, did the authors ask the subjects about usage of medication against metabolic syndromes such as antihypertensives, oral diabetes drugs or insulins? These need to be included as covariates in analyses, if not described as limitations.

 Response: Based on specific criteria proposed by the American Heart Association and the National Heart, Lung, and Blood Institute (2005), taking medication for hypertension and DM was included in diagnostic criteria for hypertension and high fasting glucose (pages 3, line 102~106).

Reviewer 2 Report

Comments: 1. Abstract: Please provide the range of age of postmenopausal women. 2. Abstract: As Abstract stand alone, author is advised to add BMI values for the overweight and obese adults. 3. What are the metabolic abnormalities included in this study? Name few in the Abstract. 4. Seems subtitles number is missing in the Results part. 5. If it’s possible, authors could categorize the women based on their age (below 65 and above 65 years). Such age division may better explain the vulnerable age group of women to CVD with or without metabolic abnormalities. 6. The main finding of this study is that 3 or more metabolic abnormalities with obesity are associated with incidence of CVD. What is the novelty in this finding? Does it differ from previous studies or differ from other ethnic groups? This part can be discussed with key metabolic abnormalities by providing their values or cutoff point. (Lines 223-131) 7. Author may provide ranges of cholesterol, TG, HDL and LDL levels in two age groups of women. 8. It would be interesting if authors can provide the lipid profile data for under weight, normal weight, overweight and obese women. 9. What is the role of physical activity among the selected population? Author can provide the physical activity data and discuss accordingly.

Author Response

Title: Combined effects of metabolic abnormalities and obesity on cardiovascular diseases among Korean postmenopausal women

We appreciate the reviewers for your valuable comments. The following are our responses to each comment. To make it easier for you to follow how changes were made, our changes appear in red ink in the revised text.

Reviewer 2.

Abstract

  1. Please provide the range of age of postmenopausal women.

Response: I added range of age among participants (page 1, line 10).

  1. As Abstract stand alone, author is advised to add BMI values for the overweight and obese adults
  2. Response: I added BMI values for the overweight and obese adults in abstract section (page 1, line 15~16).

Results

  1. Seems subtitles number is missing in the Results part

Response: I added subtitles number in result section (page 4, line 158, and 169).

  1. If it’s possible, authors could categorize the women based on their age (below 65 and above 65 years) Such age division may better explain the vulnerable age group of women to CVD with or without metabolic abnormalities.

Response: Thank you for comments. However, in this study, I want to verify the combined effect of metabolic abnormalities and obesity with controlling covariates in Korean postmenopausal women. In this, age was controlled as a covariate. However, usually aging is significant associated factor of metabolic syndrome. Thus, further studies are required to compare the combined effect of metabolic abnormalities and obesity according to age. I added need of the further studies in limitation section (page 8, line 249~251).

  1. The main finding of this study is that 3 or more metabolic abnormalities with obesity are associated with incidence of CVD. What is the novelty in this finding? Does it differ from previous studies or differ from other ethnic groups? This part can be discussed with key metabolic abnormalities by providing their values or cut-off point. (Lines 223-131)

Response: In this study, I want to verify effects of metabolic abnormalities and obesity on CVDs among Korean postmenopausal women. In previous studies, their effects on CVDs was inconsistent according to criteria for metabolic abnormalities, gender and ethnicity. In this context, this study confirmed the potential effects of metabolic abnormalities (e.g. metabolic syndrome) and obesity on the development of CVDs in Korean postmenopausal women. Through this, the evidence for emphasizing the need for management of metabolic syndrome in the prevention of CVDs among Korean postmenopausal women has been strengthened. In addition, I agreed with identification of key metabolic abnormalities associated with CVDs. However, with considering purpose of this study, further study should be performed with another purpose and hypothesis. Thus, I added need of further studies for identification of key metabolic abnormalities associated with CVDs in Korean (page 8, line 256~259).

  1. Author may provide ranges of cholesterol, TG, HDL and LDL levels in two age groups of women.

Response: As I mentioned already, comparing the combined effect of metabolic abnormalities and obesity according to age was not purpose of this study. Age was just a covariate in this study. Thus, I added range of cholesterol, TG, HDL and LDL in participants regardless of age (Table 1).

  1. It would be interesting if authors can provide the lipid profile data for underweight, normal weight, overweight and obese women

Response: I added range of cholesterol, TG, HDL and LDL according to adiposity in participants (Table 1).

  1. What is the role of physical activity among the selected population? Author can provide the physical activity data and discuss accordingly

Response: Physical activity was just controlled as a covariate. Thus, I did not discuss about results regarding physical activity.

Reviewer 3 Report

Comments to Authors 

            The current study suggest that obesity may be not a significant factor associated with CVDs among Korean postmenopausal women.

         Authors are kindly requested to emphasize the current concepts about these issues in the context of recent knowledge and the available literature. This article should be quoted in the References list.

References

  1. Association between obesity type and obstructive coronary artery disease in stable symptomatic postmenopausal women: data from the KoRean wOmen'S chest pain rEgistry (KoROSE). Menopause. 2019; 26 (11): 1272-1276. doi:10.1097/GME.0000000000001392.
  2. Beneficial Effects of Breastfeeding on the Prevention of Metabolic Syndrome Among Postmenopausal Women. Asian Nurs Res (Korean Soc Nurs Sci). 2020; 14 (3): 173-177. doi:10.1016/j.anr.2020.07.003.
  3. Association of the Risk of Osteoarthritis and Hypertension in the Korean Adult Population Aged 40-59 in Pre- and Postmenopausal Women: Using Korea National Health and Nutrition Examination Survey 2012-2016 Data. Int J Hypertens. 2021;2021:8065838. Published 2021 Feb 23. doi:10.1155/2021/8065838.
  4. Kim MH, Song MR. Association between hyperuricemia and metabolic risk components in Korean women [published online ahead of print, 2021 Jun 14]. Health Care Women Int. 2021;1-11. doi:10.1080/07399332.2021.1883025.

Author Response

Title: Combined effects of metabolic abnormalities and obesity on cardiovascular diseases among Korean postmenopausal women

We appreciate the reviewers for your valuable comments. The following are our responses to each comment. To make it easier for you to follow how changes were made, our changes appear in red ink in the revised text.

The current study suggest that obesity may be not a significant factor associated with CVDs among Korean postmenopausal women.

         Authors are kindly requested to emphasize the current concepts about these issues in the context of recent knowledge and the available literature. This article should be quoted in the References list.

References

  1. Association between obesity type and obstructive coronary artery disease in stable symptomatic postmenopausal women: data from the KoRean wOmen'S chest pain rEgistry (KoROSE). Menopause. 2019; 26 (11): 1272-1276. doi:10.1097/GME.0000000000001392.
  2. Beneficial Effects of Breastfeeding on the Prevention of Metabolic Syndrome Among Postmenopausal Women. Asian Nurs Res (Korean Soc Nurs Sci). 2020; 14 (3): 173-177. doi:10.1016/j.anr.2020.07.003.
  3. Association of the Risk of Osteoarthritis and Hypertension in the Korean Adult Population Aged 40-59 in Pre- and Postmenopausal Women: Using Korea National Health and Nutrition Examination Survey 2012-2016 Data. Int J Hypertens. 2021;2021:8065838. Published 2021 Feb 23. doi:10.1155/2021/8065838.
  4. Kim MH, Song MR. Association between hyperuricemia and metabolic risk components in Korean women [published online ahead of print, 2021 Jun 14]. Health Care Women Int. 2021;1-11. doi:10.1080/07399332.2021.1883025.

Response: I cited the suggested meaningful papers in this manuscript (Page 7, line 211 / Page 8, line 216~218).